# Measuring Compound Soil Erosion by Wind and Water in the Eastern Agro–Pastoral Ecotone of Northern China

**Degen Lin** [1,2,3,4], **Peijun Shi** [3,4,5,6,*], **Michael Meadows** [2,7,8,*], **Huiming Yang** [3,8,9], **Jing'ai Wang** [2,3,4,5,6], **Gangfeng Zhang** [3,6] and **Zhenhua Hu** [1]

1   Business School International Department, Wenzhou University, Wenzhou 325035, China; lindegen@163.com (D.L.); ying6409@126.com (Z.H.)
2   College of Geography and Environmental Sciences, Zhejiang Normal University, Jinhua 321004, China; jwang@bnu.edu.cn
3   State Key Laboratory of Earth Surface Processes and Resource Ecology, Beijing Normal University, Beijing 100875, China; hmyang09@163.com (H.Y.); zhanggf15@foxmail.com (G.Z.)
4   Academy of Plateau Science Sustainability, The People's Government of Qinghai Province—Beijing Normal University, Xining 810008, China
5   Key Laboratory of Environmental Change and Natural Disaster, MOE, Beijing Normal University, Beijing 100875, China
6   Academy of Disaster Reduction and Emergency Management, Ministry of Emergency Management & Ministry of Education, Beijing 100875, China
7   Department of Environmental & Geographical Science, University of Cape Town, Rondebosch, Cape Town 7700, South Africa
8   School of Geography and Ocean Sciences, Nanjing University, Nanjing 210093, China
9   School of Land and Tourism, Luoyang Normal University, Luoyang 471934, China
*   Correspondence: spj@bnu.edu.cn (P.S.); michael.meadows@uct.ac.za (M.M.)

**Abstract:** Land degradation induced by soil erosion is widespread in semiarid regions globally and is common in the agro–pastoral ecotone of northern China. Most researchers identify soil erosion by wind and water as independent processes, and there is a lack of research regarding the relative contributions of wind and water erosion and the interactions between them in what is referred to here as compound soil erosion (CSE). CSE may occur in situations where wind more effectively erodes a surface already subject to water erosion, where rainfall impacts a surface previously exposed by wind erosion, or where material already deposited by wind is subject to water erosion. In this paper, we use the Chinese Soil Loss Equation (CSLE) and the Revised Wind Erosion Equation (RWEQ) to calculate the rate of soil erosion and map the distribution of three types of soil erosion classified as (i) wind (wind-erod), (ii) water (water-erod), and (iii) CSE (CSE-erod) for the study area that spans more than 400,000 km$^2$ of sand- and loess-covered northern China. According to minimum threshold values for mild erosion, we identify water-erod, wind-erod, and CSE-erod land as occurring across 41.41%, 13.39%, and 27.69% of the total area, while mean soil erosion rates for water-erod, wind-erod, and CSE-erod land were calculated as 6877.65 t km$^{-2}$ yr$^{-1}$, 1481.47 t km$^{-2}$ yr$^{-1}$, and 5989.49 t km$^{-2}$ yr$^{-1}$, respectively. Land subject to CSE-erod is predominantly distributed around the margins of those areas that experience wind erosion and water erosion independently. The CSLE and RWEQ do not facilitate a direct assessment of the interactions between wind and water erosion, so we use these equations here only to derive estimates of the relative contributions of wind erosion and water erosion to total soil erosion and the actual mechanisms controlling the interactions between wind and water erosion require further field investigation. It is concluded that CSE is an important but underappreciated process in semiarid regions and needs to be accounted for in land degradation assessments as it has substantial impacts on agricultural productivity and sustainable development in regions with sandy and/or loess-covered surfaces.

**Keywords:** soil erosion rates; regional soil erosion map; land degradation; CSLE model; model simulation

## 1. Introduction

Land degradation induced by soil erosion is prominent in semiarid regions globally [1–5]. The agro–pastoral ecotone of northern China has characteristics that are typical of a semiarid climate transition zone in that the ecological environment of the region is fragile, and inappropriate land-use management has resulted in severe accelerated soil erosion in the region [6–8]. Soil erosion types commonly include erosion by water, wind, freeze-thaw processes, and combinations of processes in what we refer to here as compound soil erosion (CSE). Interactions between aeolian and fluvial systems have been studied at various scales [9,10] but there is little information on their relative importance and how they may interact to affect accelerated soil erosion. CSE occurs when wind and water erosion processes interact and the mutual effects of these erosive forces are combined. This ensues, for example, when wind acts on a surface already affected by water erosion/deposition, or when rainfall impacts a surface previously exposed by wind erosion/deposition. CSE typically occurs in landscapes where erosion gullies are filled by wind-blown sediments, or where gullies develop on wind-eroded surfaces; such situations are common in the typical sandy and loess-covered areas of northern China. Indeed, many sand- and loess-covered areas are influenced by both wind and water erosion processes, and CSE must therefore be considered as a potentially important component of land degradation in semiarid regions [11].

Situations whereby wind and water erosion processes act in concert with each other occur commonly in semiarid regions, especially on windward slopes in hilly topography and in association with gullies [12,13]. Several studies have suggested that CSE intensity is increasingly important as an element of landscape evolution, both in China [7,14] and elsewhere [8,15–17]. However, processes whereby wind and water erosion act together are highly complex and have not so far been adequately accounted for in terms of distribution or impact [8]. Thus, measurement and mapping of the scale and distribution of CSE are needed to improve the reliability of estimates of soil erosion and land degradation in vulnerable semiarid regions.

Studies of soil erosion in ecological transition zones tend to consider water erosion [18,19] or wind erosion [20–23] as spatially and temporally independent processes. A small number of studies, however, have indicated that wind and water erosion processes may occur at the same locality sequentially, or even overlapping in time, and therefore influence the rate of soil loss [14,24–26]. Zou [14], for example, describes CSE as characteristic in more than 5% of the region studied and has defined quantitatively the 'CSE belt' in northern China, while Yang [25] also identified the distribution of "cross-erosion" (in Chinese, this is equivalent to CSE as used here). In some studies, CSE is suggested to augment the rate of soil loss compared to other areas [7,14], and this has also been demonstrated experimentally in laboratory simulation studies [27–30]. Tuo et al. [27] found that the relative proportions of different grain sizes were different in places where CSE occurred, while Yang et al. [28] reported that wind actually subdued water erosion on slopes under moderate rainfall intensities. In summary, most soil erosion studies conducted to date have focused independently on wind or water erosion, and few have taken into account the compound effects of both processes acting in concert [8,31].

Previous studies by the present research group have focused on a number of aspects of soil erosion in northern China, including modeling [32], quantification of the impacts of wind speed variability [33], experimental studies using a wind tunnel and a rainfall simulator [28], and a pilot study of the combined influence of wind and water erosion [29]. This paper aims at developing a more refined understanding of the mechanisms of CSE and—in deploying the Chinese Soil Loss Equation (CSLE) [34,35], which has been developed for estimating annual soil erosion by water in China, and the Revised Wind Erosion equation (RWEQ) [36], which has been developed as a tool for identifying practices that control windblown soil loss—to map the distribution and rates of erosion in a selected study area of different soil erosion types, including CSE. The study provides valuable theoretical insights into the interaction between aeolian and fluvial processes and the nature of integrated soil

erosion processes in general. In so doing we offer practical insights for improving soil conservation in an important food production area of China.

This aim is achieved through the following objectives: (i) To apply standard models of soil erosion to calculate the intensity of wind erosion, water erosion, and wind erosion and water erosion combined. (ii) To map the distribution of the three erosion types. (iii) In order to assess the accuracy of the classification of the three erosion types, 18 sites representing the environmental characteristics across the study area were selected for ground-truthing. At each of these sites, field investigations included detailed observations of local conditions which were recorded photographically. (iv) To compare the results with those of other relevant soil erosion results.

## 2. Materials and Methods

### 2.1. Study Area

The research was conducted in the agro–pastoral ecotone of northern China (Figure 1) which is identical to that studied by Lin et al. [32]. The study area (107°0′0″ E–125°0′0″ E, 38°0′0″ N–48°0′0″ N) includes 84 counties in the Inner Mongolia Autonomous Region, Liaoning Province, Heilongjiang Province, Jilin Province, and Hebei Province, encompassing a total area of $4.40 \times 10^5$ km$^2$. The area falls at the margins of the monsoon zone with a mean annual precipitation of 250–500 mm, which, along with 25–50% interannual precipitation variability, restricts agriculture to irrigated crop production and pasture. In general, the topography is undulating which, coupled with low rainfall and disturbance following replacement of natural vegetation, has promoted soil degradation in the form of water erosion, wind erosion, and CSE.

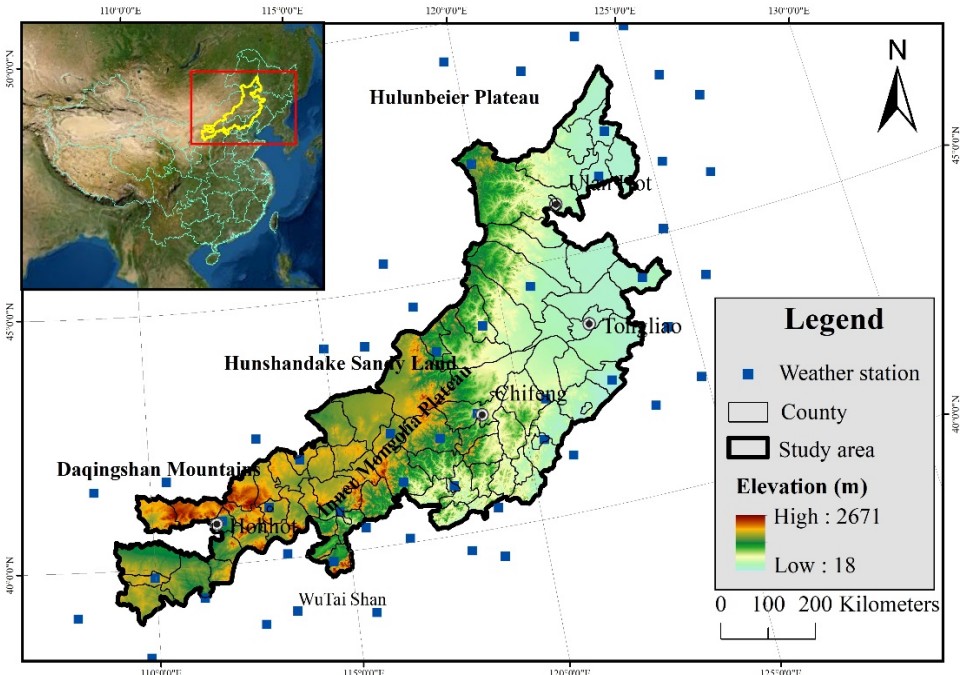

**Figure 1.** Study area.

There are great differences in the types of landforms in the study area. The northern end is the Hulunbeier Plateau (650–750 m), the southeast is the transition zone between the Northeast Plain and the Inner Mongolia Plateau, namely, Horqin Sandy Land (200–700 m), the northwest is the southern margin of the Inner Mongolia Plateau (1300–1800 m), and the southwest is Daqingshan Mountains (1400–2000 m). There are two main soil types according to the characteristics of parent material at the surface, viz., aeolian sand and loess. Among them, the Northwest Hulunbuir Plateau, Horqin, and Hunshandake Sandy Land are covered by immature sandy soils of aeolian origin, while loess is characterized

by brown and chestnut soils [37]. The study area lies in the transition zone between forest and grassland, much of which has been degraded due to inappropriate management practices [37]. In 2010, arable land accounted for 31.01%, forest land accounted for 16.51%, grassland accounted for 39.63%, water bodies 2.06%, construction land 2.46%, and unused land (including sandy land) accounted for 7.86% [38].

*2.2. Data*

A wide range of relevant data were obtained from a variety of primary and secondary sources as indicated in Table 1. These include meteorological, land use, soil, and vegetation cover data, as well as a digital elevation model (DEM) at 30 m resolution from which slope characteristics were obtained. We resampled the resolution (30 m × 30 m) of all original data using ArcGIS software 10.5.

**Table 1.** List of data used in the study.

| Type of Data | Data Name | Temporal Resolution | Spatial Resolution | Source |
| --- | --- | --- | --- | --- |
| Meteorological data | DS3505—Surface Data Hourly Global | 3 h, 1961–2015 | Weather station | NOAA, http://gis.ncdc.noaa.gov/map/viewer/#app=clim&cfg=cdo&theme=hourly&layers=1&node=gis (accessed on 17 May 2022) |
| | China surface climate data daily data set (V3.0) | Daily, 1961–2015 | Weather station | China Meteorological Data Service Center (CMDC), http://data.cma.cn/ (accessed on 17 May 2022) |
| Land use | China Land Use Status Remote Sensing Monitoring Database | 1990, 1995, 2000, 2005 and 2010 | 1 km × 1 km | Resource and Environment Data Cloud Platform, http://www.resdc.cn (accessed on 17 May 2022) |
| | Landsat TM images | 2000, 2005, 2010 and 2015 | 30 m × 30 m | USGS. http://earthexplorer.usgs.gov/ (accessed on 17 May 2022) |
| DEM | Land elevation data | 2005 | 30 m × 30 m | ASTER GDEM, https://earthdata.nasa.gov/ (accessed on 17 May 2022) |
| Soil humidity | This global ECV soil moisture data set | 1978–2014, monthly | 0.25° × 0.25° | European Space Agency, http://www.esa-soilmoisture-cci.org/node/145 (accessed on 17 May 2022) |
| Snow cover | China Snow Deep Time Series Dataset | 1978–2014, daily | 1 km × 1 km | National Cryosphere Desert Data Center. http://www.cryosphere.csdb.cn/portal/metadata/d9a9e8ae-3e8f-4c1a-b1a6-a739a405c971 (accessed on 17 May 2022) |
| Soil | Soil and Terrain Database (SOTER) for China | 2014 | Vector unit | World soil information (ISRIC), https://files.isric.org/public/soter/CN-SOTER.zip (accessed on 17 May 2022) |
| Survey data | The vegetation Coverage in Eastern Section of Farming–Pasture Ecotone of Northern China | 2015–2016 | 1 m × 1 m | Research group of "research on land use and compound soil erosion by wind and water in study area" |
| Erosion verification data | Spatial Distribution Data of Soil Erosion in China | 1995 | 1 km × 1 km | Resource and Environment Data Cloud Platform, http://www.resdc.cn (accessed on 17 May 2022) |
| | China Soil and Water Conservation Bulletin | 2004–2015 | Runoff Plots | Ministry of Water Resources, http://www.swcc.org.cn/CountryTopics.asp (accessed on 17 May 2022) |

*2.3. Methodology*

2.3.1. Water Erosion Model

Liu et al. [35] developed the Chinese Soil Loss Equation (CSLE) based on a refinement of the Universal Soil Loss Equation (USLE) and Revised Universal Soil Loss Equation (RUSLE) [39]. The water erosion model used here is based on the Chinese water erosion forecasting model, which offers a simple and practical means of estimating soil erosion by water and has been widely applied [40,41] in different regions of the country (Equation (1)). The basic function is:

$$A = R \times K \times L \times S \times B \times E \times T, \tag{1}$$

where $A$ is the calculated mean spatial and temporal soil loss per unit area over time, expressed in tonnes per square kilometer per year, $t\,km^{-2}\,yr^{-1}$; $R$ = rainfall-runoff erosivity factor, viz., the rainfall erosion index plus a factor for any significant runoff from snow melt, $MJ\,mm\,ha^{-1}\,h^{-1}$ per year; $K$ is the soil erodibility factor, $t\,ha^{-1}\,MJ\,mm^{-1}$; $S$ and $L$ are dimensionless slope steepness and slope length factors; $B$, $E$, and $T$ are dimensionless factors of biological-control, engineering-control, and tillage practices, respectively. Biological-control (essentially the vegetation cover factor) was calculated using the method of Lin et al. [32]. Engineering-control ($E$), in the form of soil conservation management interventions, was implemented on a large scale in China in 1993, although most soil and water conservation projects were introduced beyond the study area [42] and so for the purposes of this study, we assume no engineering control factor. Liu et al. [35] provide tillage practice factors ($T$) for cultivated land under different slopes, as shown in Table 2.

**Table 2.** The T value of tillage measures in different slope areas in the study area.

| Slope(°) | ≤1 | 1–3 | 3–9 | 9–13 | 13–17 | 17–21 | 21–25 | ≥25 |
|---|---|---|---|---|---|---|---|---|
| Tillage | 0.74 | 0.59 | 0.60 | 0.62 | 0.68 | 0.75 | 0.81 | 0.92 |

The formulas for calculating the various water erosion factors are shown in Table 3. We used the data from Table 1 to calculate erosion factors by using the Equations (2)–(15) in Table 3.

**Table 3.** Formulas for calculating various factors of water erosion equation.

| Factor | Formulas | | References |
|---|---|---|---|
| Rainfall erosivity (R) | $\beta = 0.8363 + \frac{18.144}{P_{(d12)}} + \frac{24.455}{P_{(d12)}}$ | (2) | [43] |
| | $\alpha = 21.586\beta^{-7.1891}$ | (3) | |
| | $R_{halfm} = \alpha \sum\limits_{k=1}^{m} (P_k)^{\beta}$ | (4) | |
| | $R_{year} = \sum\limits_{i=1}^{24} R_{halfmi}$ | (5) | |
| | $\overline{R} = \frac{1}{n} \sum\limits_{i=1}^{n} R_{yeari}$ | (6) | |
| Topographic (LS) | $LS = \left(\frac{A_s}{22.13}\right)^m \left(\frac{\sin(\theta)}{0.0896}\right)^n$ | (7) | [44] |
| Vegetation cover (B) | $B = C_s \cdot C_c$ | (8) | [32] |
| | $C_c = 1 - (0.01V_c + 0.0859)\exp(-0.0033h)$ | (9) | |
| | $C_c = 0.988\exp(-0.11V_c)$ | (10) | |
| | $C_s = 1.029 * \exp(-0.0235V_r)$ | (11) | |
| | $C_s = \exp(-0.0206V_r)$ | (12) | |
| | $C = (V_{cr}C_{cr} + V_{gr}C_{gr} + V_{fo}C_{fo})/(V_{cr} + V_{gr} + V_{fo})$ | (13) | |
| Soil erodibility (K) | $K = 0.0364 - 0.0013[ln(\frac{OM}{D_g}) - 5.6706]^2 - 0.015exp[-28.9589(log(D_g) + 1.827)]^2$ | (14) | [45] |
| | $D_g = exp\left(0.01 \times \sum \sum\limits_{i=1}^{n} f_i lnm_i\right)$ | (15) | |

This is the explanation for the terms in equations of the Table 3. Four steps are involved in calculate water erosion calculation, viz., the calculation of rainfall erosivity factor (R) where $R_{year}$ is the annual rainfall erosion force, MJ·mm/(ha·h·a); $\overline{R}$ is the multi-year average annual rainfall erosion force, MJ·mm/(ha·h·a); $R_{halfm}$ is the value of rainfall erosion in a half-month period, MJ·mm/(ha·h); $m$ is the number of days in the half-month period; $P_k$ is the daily rainfall (mm) on the $k$-th day in the half-month period; $P_{(d12)}$ is the multiyear average daily rainfall (mm) with a daily rainfall excess of 12 mm; and $P_{(y12)}$ is multiyear average annual rainfall (mm) with a daily rainfall excess of 12 mm. Secondly, we calculated the topographic factor (LS), where $A_s$ is the unit contribution area (m); $\theta$ is the slope, expressed in radians; and $m$ and $n$ are exponents, with values ranging from 0.4–0.56 and 1.2–1.3, respectively. Thirdly, we calculated the vegetation cover factor (B), where $C_c$ and $C_s$ are the canopy cover factor and the surface cover factor, respectively; $V_c$ and $V_r$ are the canopy cover (%) and surface cover (%), respectively; and $h$ is the canopy height (cm). Based on the C factor algorithm of the standard plot, the regional C factor algorithm is proposed, as shown in Equation (13), where $C$ is the C factor of the km-sized grid, $V_{cr}$ is the cropland coverage (%), $V_{gr}$ is the grassland coverage (%), and $V_{fo}$ is the forest coverage (%). $C_{cr}$ is the vegetation cover factor of cropland, $C_{gr}$ is the grassland vegetation cover factor, and $C_{fo}$ is the vegetation cover factor of woodland. Fourthly, we calculated the soil erodibility factor (K). $K$ is the soil erodibility (t·hm$^2$·h/(hm$^2$·MJ·mm); $D_g$ is the geometric mean diameter particle of the soil (mm); $OM$ is the content of soil organic carbon (%); $f_i$ is the $i$-th particle, $m_i$ is the arithmetic mean value (mm) of the particle size of the $i$-th particle, $n$ is the numerical classification of particle size, including $SAN$, $SIL$, and $CLA$. $SAN$ is sand content (%); $SIL$ is the silt content (%); and $CLA$ is clay content (%).

### 2.3.2. Wind Erosion Model

The Revised Wind Erosion Equation (RWEQ) [46] is frequently used for wind erosion simulation and is regarded as a suitable tool for the large-scale prediction of wind erosion potential [47]. The RWEQ uses a number of input factors to estimate the maximum transport capacity ($Q_{max}$) (Equation (16)). Critical field length (S) is determined when 63.2% of $Q_{max}$ is attained (Equation (17)), and soil loss (SL) is the total transport capacity (Equation (18)).

$$Q_{max} = 109.8 \cdot \left[ WF \cdot EF \cdot SCF \cdot K' \cdot C \right], \tag{16}$$

$$S = 150.71 (WF \cdot EF \cdot SCF \cdot K' \cdot C)^{-0.3711}, \tag{17}$$

$$S_L = \frac{2z}{S^2} Q_{max} e^{-(z/s)^2}, \tag{18}$$

where $Q_{max}$ is the maximum transport capacity (kg/m); $S$ is the length of the key plot (m); $S_L$ is soil erosion per unit area (kg m$^{-2}$), and 1 kg m$^{-2}$ can convert to 1000 t km$^{-2}$ yr$^{-1}$ per year; $z$ is the calculated downwind distance (m), and the value of this calculation is 50 m; $WF$ is the wind factor (kg/m); $EF$ is the erodible fraction (proportion of particles below 0.84 mm in diameter); $SCF$ is a dimensionless surface crust factor; $K'$ is a dimensionless random soil surface roughness factor; and $C$ is a dimensionless comprehensive wind erosion factor.

The formulas for calculating the various wind erosion factors are shown in Table 4. We used the data from Table 1 to calculate erosion factors by using the Equations (19)–(26) in Table 4.

This is the explanation for the terms in equations of the Table 4. Four steps were used to calculate wind erosion. Firstly, we calculated wind erosion climate (WF), where $WF$ is the wind erosion climatic factor (kg/m); $W_f$ is the wind factor (m/s)$^3$; $SW_f$ is the soil moisture factor (0–1), dimensionless; and $SD$ is the snow cover factor, dimensionless. $U_2$ is the wind speed two meters above the ground (m/s); $U_t$ is the critical wind speed, generally set to 5 m/s; $N_d$ is the time period for measuring the wind speed, generally about 15 days; $N$ is the number of wind speed observations (generally no less than 500 times); and $P$ (*Snow_depth* > 25.4 mm) is the probability that the snow cover depth in the calculation period is greater than 25.4 mm, and the calculation period is 15 days. Secondly, we

calculated surface roughness (K′). $K_r$ is the terrain roughness (cm); $C_{rr}$ is the random roughness factor (cm); $L$ is the relief parameter; and $\Delta H$ is the altitude difference within the distance $L$. Thirdly, we calculated comprehensive vegetation (C), where SC is the vegetation cover (%). Fourthly, soil erodibility (EF) and soil crust (SCF) factors were calculated, where $Sa$ is the sand content of soil (5.5%~93.6%); $Si$ is the silt content (0.5%~69.5%); $Sa/Cl$ is the ratio of sand to clay (1.2%~53.0%); $Cl$ is the clay content (5.0%~39.3%); $CaCO_3$ is calcium carbonate content (0~25.2%); and $OM$ is the organic matter content (0.32%~4.74%).

**Table 4.** Formulas for calculating various factors of wind erosion equation.

| Factor | Formulas | | References |
|---|---|---|---|
| Wind erosion climate (WF) | $W_f = \dfrac{\sum\limits_{i=1}^{n} U_2(U_2 - U_t)^2}{N} \times N_d$ | (19) | [46] |
| | $SW_f = 1 - SW$ | (20) | |
| | $SD = 1 - P(Snow\_depth > 25.4 \text{ mm})$ | (21) | |
| Surface roughness (K′) | $K' = e^{(1.86K_r - 2.41K_r^{0.934} - 0.127C_{rr})}$ | (22) | [48] |
| | $K_r = 0.2 \times \dfrac{\Delta(H)^2}{L}$ | (23) | |
| Comprehensive vegetation (C) | $C = e^{-0.0438SC}$ | (24) | [49] |
| Soil erodibility (EF) | $EF = \dfrac{29.09 + 0.31Sa + 0.17Si + 0.33\frac{Sa}{Cl} - 2.59OM - 0.95CaCO_3}{100}$ | (25) | [46] |
| Soil crust (SCF) | $SCF = \dfrac{1}{1 + 0.0066(Cl)^2 + 0.021(OM)^2}$ | (26) | [46] |

### 2.3.3. Estimation of CSE

According to the characteristics of the surface in question, CSE can be classified into wind and water sequential erosion (WWSE) and wind and water combined erosion (WWCE). WWCE refers to the simultaneous occurrence of wind and water erosion, whereby the coupling effect alters the erosive ability and makes the compound erosion force different from when it applies only through one type of process. In WWSE, erosive forces act alternately, whereby one erosive force transports and deposits surface materials and prepares the material for reactivation by the other force. In this case, the overall erosion force may also be changed.

WWSE can be calculated using wind erosion and water rates as shown in Equation (27).

$$SE_1 = W_a + W_i, \tag{27}$$

where $SE_1$ is the WWSE rate, t km$^{-2}$ yr$^{-1}$, $W_a$ is the water erosion rate (t km$^{-2}$ yr$^{-1}$); and $W_i$ is the wind erosion rate (t km$^{-2}$ yr$^{-1}$).

WWCE per year can be calculated using Equation (28).

$$SE_2 = W_i W_a(T_1) + W_a W_i(T_2) + W_a(T_3) + W_i(T_3) \tag{28}$$

where $SE_2$ is the WWCE rate, t km$^{-2}$ yr$^{-1}$, $W_a$ is the water erosion rate (t km$^{-2}$ yr$^{-1}$), $W_i$ is the wind erosion rate (t km$^{-2}$ yr$^{-1}$); $W_i W_a$ is the effect of wind on water erosion; and $W_a W_i$ is the effect of water on wind erosion. According to the wind factor ($W_f$) and rainfall erosivity factor (R), $T_1$ is the length of time of wind impact on water erosion, and often happens in June (See Section 4.1). $T_2$ is the length of time of water impact on wind erosion, and often happens in September (See Section 4.1). $T_3$ is the time of sequential erosion.

### 2.3.4. Mapping the Distribution of Soil Erosion Types

In China, acceptable (allowable) threshold rates of soil erosion are classified as 1000 t km$^{-2}$ yr$^{-1}$ for water erosion, and as 200 t km$^{-2}$ yr$^{-1}$ for wind erosion [50–52]. We identified all the 30 m × 30 m grid squares in the study area in which these threshold values were exceeded according to the system illustrated in Figure 2. The erosion types on

the field were estimated by 1 m × 1 m field investigations [32]. We mapped the erosion types, including CSE, using ArcGIS software 10.5, accordingly. Additionally, the erosion types were validated by photographs of typical examples of wind-erod type, water-erod type, and CSE-erod type.

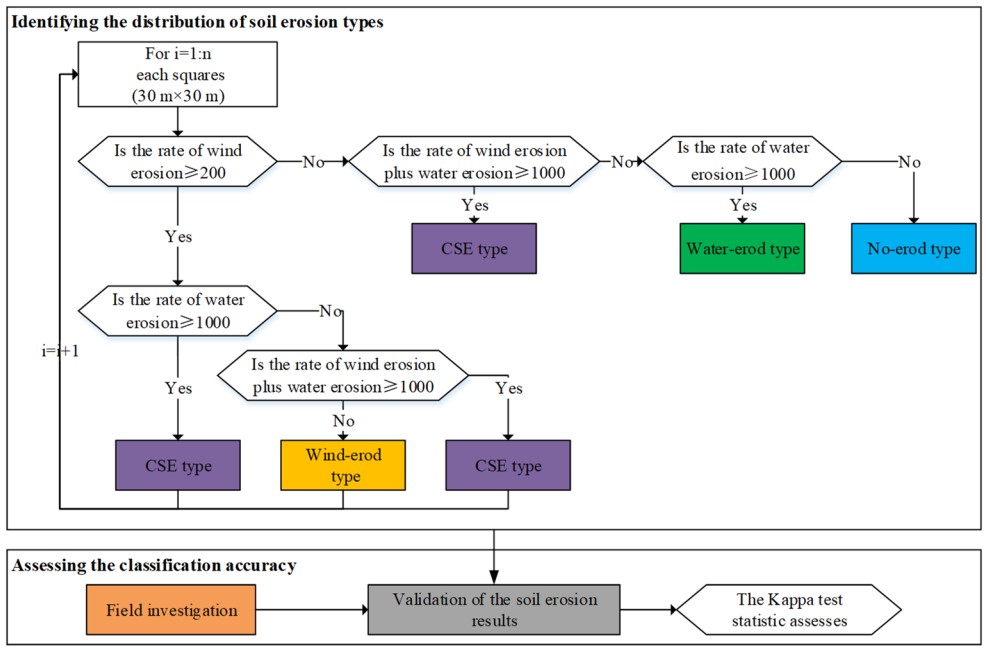

**Figure 2.** Scheme illustrating the methods used to identify the three erosion types.

2.3.5. Assessing Classification Accuracy of Soil Erosion Types

In order to assess the mapped accuracy of soil erosion types, we calculated the Kappa coefficient [53] based on a confusion matrix. The Kappa coefficient is proposed here for assessing the accuracy of classifiers. Cohen's Kappa statistic compensates for classifications that may be due to chance and is a widely used method in these circumstances [54]. The confusion matrix table is used to estimate overall accuracy ($A_o$), determined by dividing the total diagonal sum by the random number of points as in Equation (29), and this provides the validation data.

$$A_o = \frac{X}{N} \times 100\%, \tag{29}$$

where $A_o$ is overall accuracy (%); $X$ is number of diagonal values; and $N$ is number of random points.

Cohen's Kappa coefficient [55] was used to determine the accuracy of the calculation using two measurement techniques (measurement by computer and by actual survey), whereby the coefficient varies from −1 to 1 [56], as in Equation (30).

$$\text{khat} = \frac{A_o - M_t}{1 - M_t}, \tag{30}$$

where khat is the Cohen's Kappa coefficient; $A_o$ is overall accuracy (%); and $M_t$ is test matrix product.

The matrix product is the sum of the diagonal values, while the aggregate number of the matrix product is the sum of values in rows and columns as in Equation (31).

$$M_t = \frac{M_p}{M_{pcum}}, \tag{31}$$

$M_p$ is matrix product; and $M_{pcum}$ is cumulative sum of matrix product.

Accuracy was also assessed by ground-truthing during field investigation.

2.3.6. Validation of the Soil Erosion Results

Previous soil erosion studies have not systematically considered different types of CSE. Therefore, we compared two types of soil erosion maps which contain wind and water erosion, viz., (i) spatial distribution raster data of Chinese soil erosion (data from Data Center for Resources and Environmental Sciences, Chinese Academy of Sciences (RESDC), https://www.resdc.cn/data.aspx?DATAID=259, accessed on 17 May 2022), hereinafter referred to as the CAS-erosion map; and (ii) the First National Water Conservancy Census General Survey of Soil and Water Conservation: Soil Erosion, which provided by Beijing Normal University and institute of mountain, Hazards and environment CAS, hereinafter referred to as BNU-erosion map. Both maps were made using the National Soil Erosion Classification Grading Standard [57], and are therefore suitable for comparison with the results of this study.

We constructed a scatter plot with a 1:1 line and data distribution of water erosion magnitude comparing the CAS and BNU outputs with this study and then compared wind erosion magnitude between CAS and this study using the same method.

## 3. Results

### 3.1. Soil Erosion Intensity

Annual water erosion and wind erosion rates were calculated from 1975 to 2015 for each 30 m × 30 m grid square in the study area. According to the soil erosion intensity rating system of the Water and Soil Conservation Department of the Ministry of Water Resources [57], we present here the spatial results for water-erod intensity (Figure 3a), wind-erod intensity (Figure 3b), and CSE-erod intensity (Figure 3c).

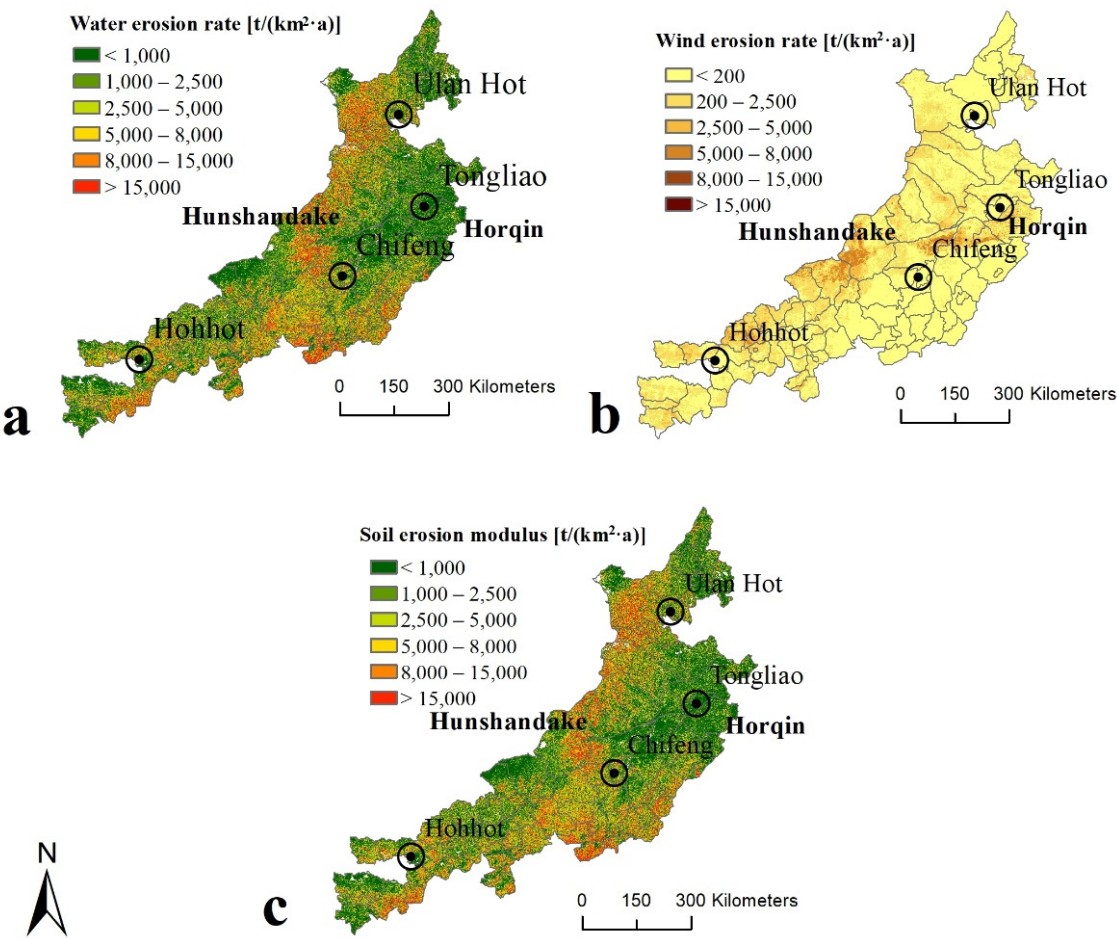

**Figure 3.** The rate of soil erosion in the study area ((**a**): water erosion; (**b**): wind erosion; (**c**): soil erosion) based on CSLE and RWEQ models.

Figure 3 and Table 5 reveal that the mean annual soil erosion rate over an erosion area of $3.49 \times 10^5$ km$^2$, which represents some 82.49% of the total study area, is 4859.96 t km$^{-2}$ yr$^{-1}$. The erosion area is 20.26% larger than when water erosion alone is considered, and 48.18% larger than for wind erosion alone. The erosion rate is 11.03% higher than in the case of water erosion alone, and 88.97% higher for wind erosion alone. Areas of severe water erosion are concentrated in the central and northern mountainous parts of the study area; the area of slight water erosion is associated with the Horqin and Hunshandake sandy lands which are, in contrast, very strongly impacted by wind erosion. Wind erosion in the western part of the study area is much more prominent than it is in the east, mainly because the western parts form the windward slopes in the path of the winter monsoon.

**Table 5.** Erosion rate and area in the study area.

| Types | Erosion Rate<br>t km$^{-2}$ yr$^{-1}$ | Erosion Area<br>km$^2$ | Proportion of the Study Area<br>% |
|---|---|---|---|
| Water erosion | 4323.67 | $2.63 \times 10^5$ | 62.23 |
| Wind erosion | 536.15 | $1.45 \times 10^5$ | 34.31 |
| Soil erosion | 4859.96 | $3.486 \times 10^5$ | 82.49 |

Note: The rate and area of soil erosion here is considered water erosion and wind erosion together, and no consideration of CSE.

### 3.2. Soil Erosion Type Distribution

According to the method of identifying the distribution of different types of soil erosion, water-erod (Figure 4a), wind-erod (Figure 4b), and CSE-erod (Figure 4c) are identified.

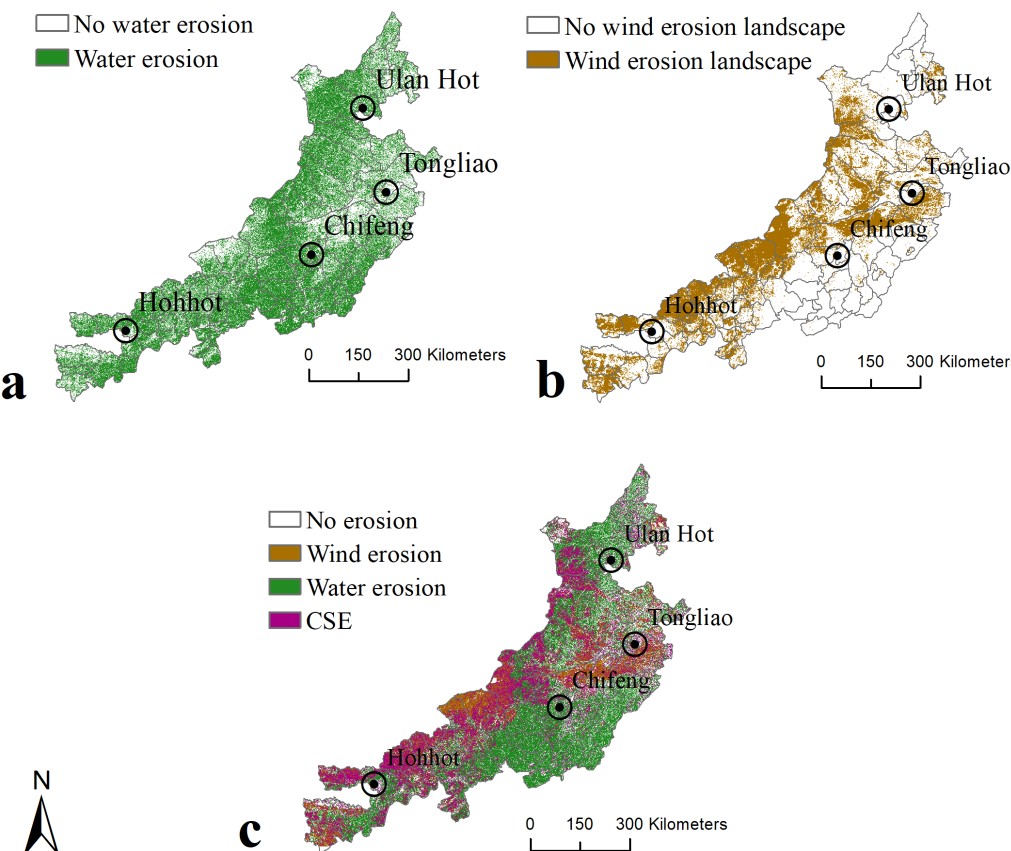

**Figure 4.** Distribution of soil erosion types in the study area ((**a**): water-erod; (**b**): wind-erod; (**c**): CSE-erod).

Figure 4 shows that water-erod, wind-erod, and CSE-erod land occurs across 41.41%, 13.39%, and 27.69% of the total area, while mean soil erosion rates for water-erod, wind-erod, and CSE land were calculated as 6877.65 t km$^{-2}$ yr$^{-1}$, 1481.47 t km$^{-2}$ yr$^{-1}$, and 5989.49 t km$^{-2}$ yr$^{-1}$, respectively. Among these, water-erod is the most widely distributed, particularly in the southeastern and northern mountainous parts of the study area, while wind-erod squares are most prominent in the Hunshandake and Horqin Sandy Lands. Land subject to CSE-erod is concentrated around the margins of those areas that experience wind erosion and water erosion independently, and it is especially concentrated on the windward slopes in the western mountainous region and in the eastern Horqin Sandy Land.

### 3.3. Validation of Water Erosion

We compared water erosion as presented in the CAS-erosion map, BNU-erosion map, and the mapped outcome of our study at the county level (Figures 5 and 6).

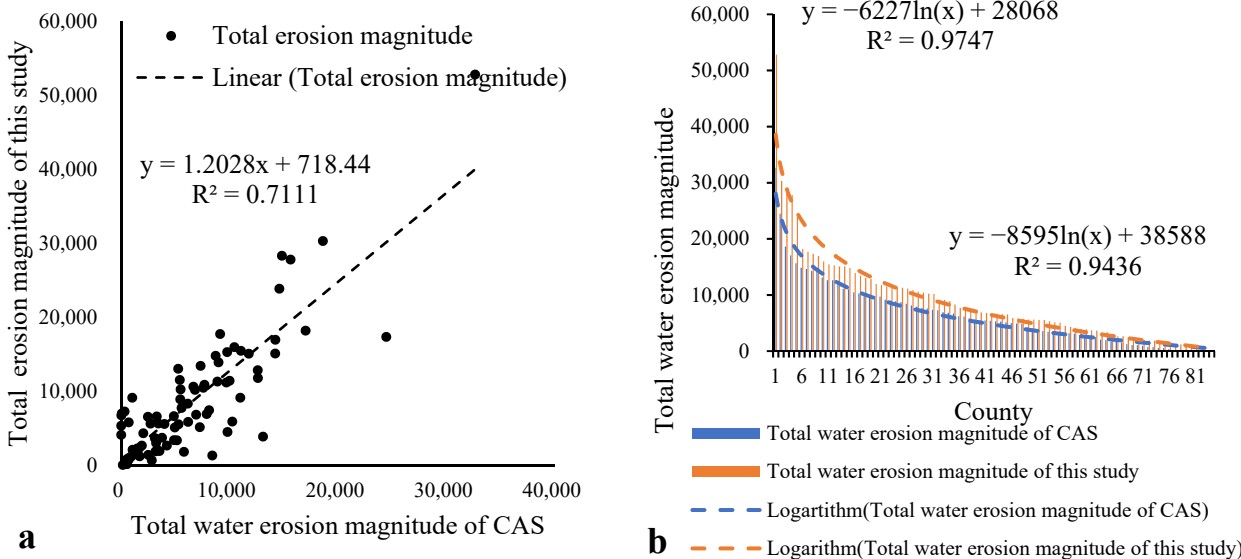

**Figure 5.** Scatter plot and data distribution graph of water erosion magnitude in the study area of between CAS and this study ((**a**): water-erod magnitude scatter plot; (**b**): water-erod data distribution).

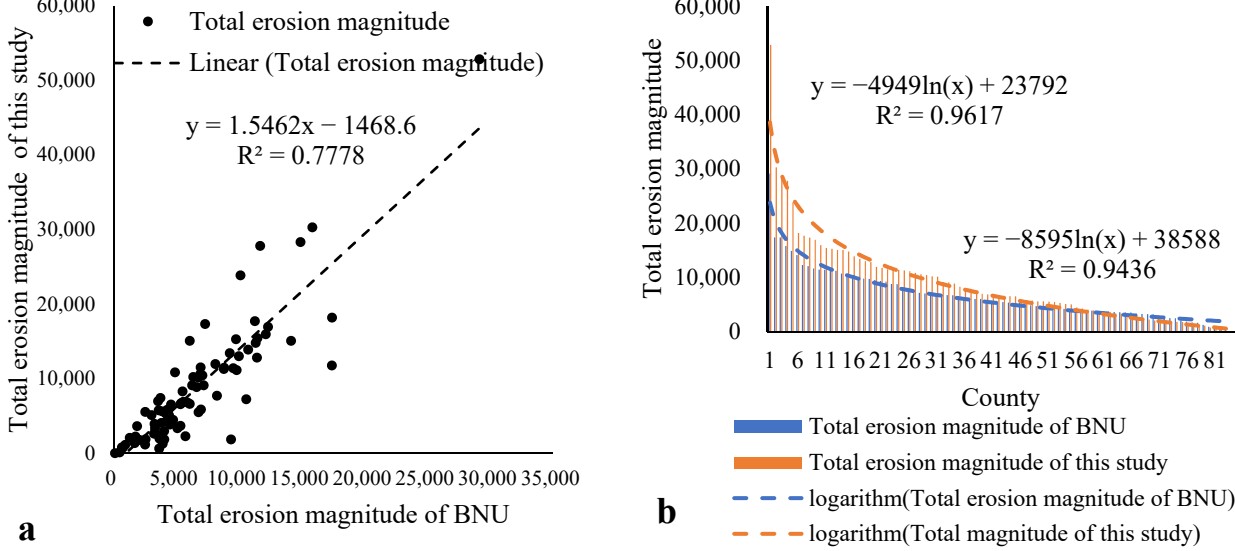

**Figure 6.** Scatter plot and data distribution of water erosion magnitude in the study area of between BNU and this study ((**a**): water-erod magnitude scatter plot; (**b**): water-erod data distribution).

According to the Wilcoxon Signed-Rank Test, the different maps for water erosion are statistically similar ($p < 0.05$). We can see that the magnitude of water erosion based on our study is basically consistent ($R^2 = 0.71$) with the CAS-erosion map (Figure 5a) and with the BNU-erosion map ($R^2 = 0.78$) (Figure 6a) at both the CAS-erosion and BNU-erosion maps (Figures 5b and 6b, respectively), although slight erosion by water appears to be underestimated. In general, however, the results of this study provide a reliable estimate of the magnitude and distribution of soil erosion and may be considered to accurately represent water erosion intensity.

### 3.4. Validation of Wind Erosion

We compared the wind erosion assessment in this study with the CAS-erosion map at the county level (Figure 7) and, although there is broad consistency between the two ($R^2 = 0.60$), the CAS map indicates somewhat higher rates overall. The main reason is that the CAS-erosion map classified the western (windward) part of the study area as wind-erod only, whereas our model simulation and field investigation reveal that this area also has a high rate of water erosion and should be classified as CSE-erod.

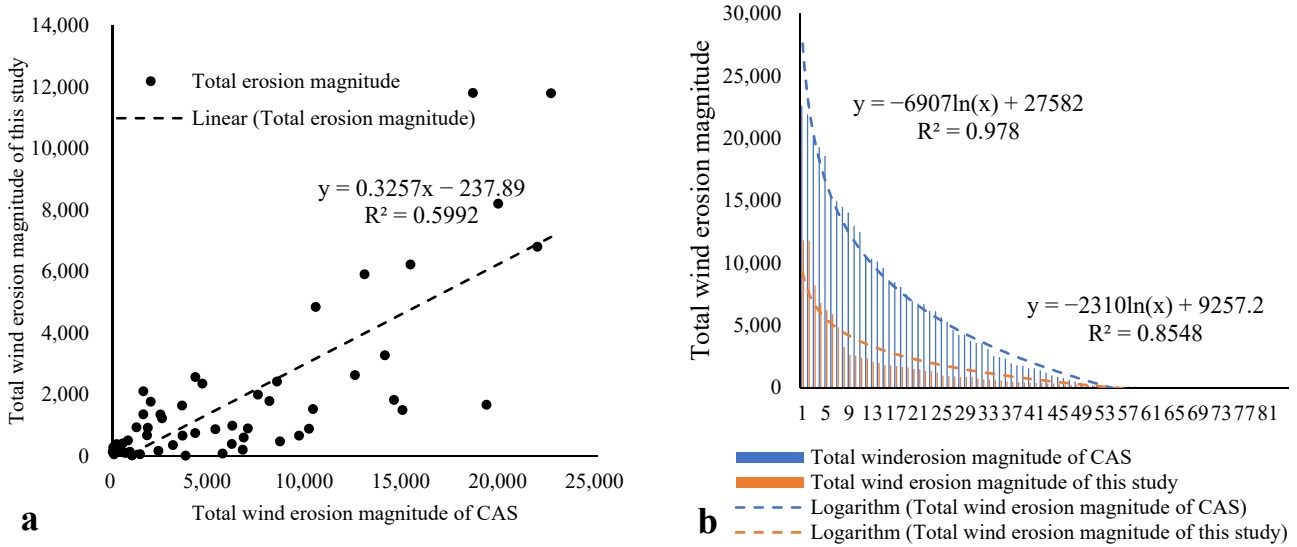

**Figure 7.** Scatter plot and data distribution of wind erosion magnitude and water erosion magnitude in the study area ((**a**): wind-erod magnitude scatter plot; (**b**): wind-erod data distribution map).

### 3.5. Validation of CSE

A field investigation of soil erosion types was conducted in order to validate the model results, and photographs of typical examples of CSE are shown in Figure 8.

Figure 8 illustrates typical examples of CSE-erod in the study area, revealing that such erosion is associated with a range of conditions, including unpaved roads, seasonally bare sloping farmland, sand covered loess ditches, loess-covered and sandy roads, loess and sand mounds, and on roads covered by blown sand; evidence of disturbance due to human activity is usually present.

Based on field observations and the accuracy assessment, Figure 8 and Table 6 reveal that accuracy for water-erod, wind-erod, and CSE types are 100%, 100%, and 66.67%, respectively. Overall, identification accuracy for soil erosion types is 72.22%. The calculated Kappa coefficient of 0.61 indicates that the classification of soil erosion types is acceptable and supported by ground-truthing during field investigation.

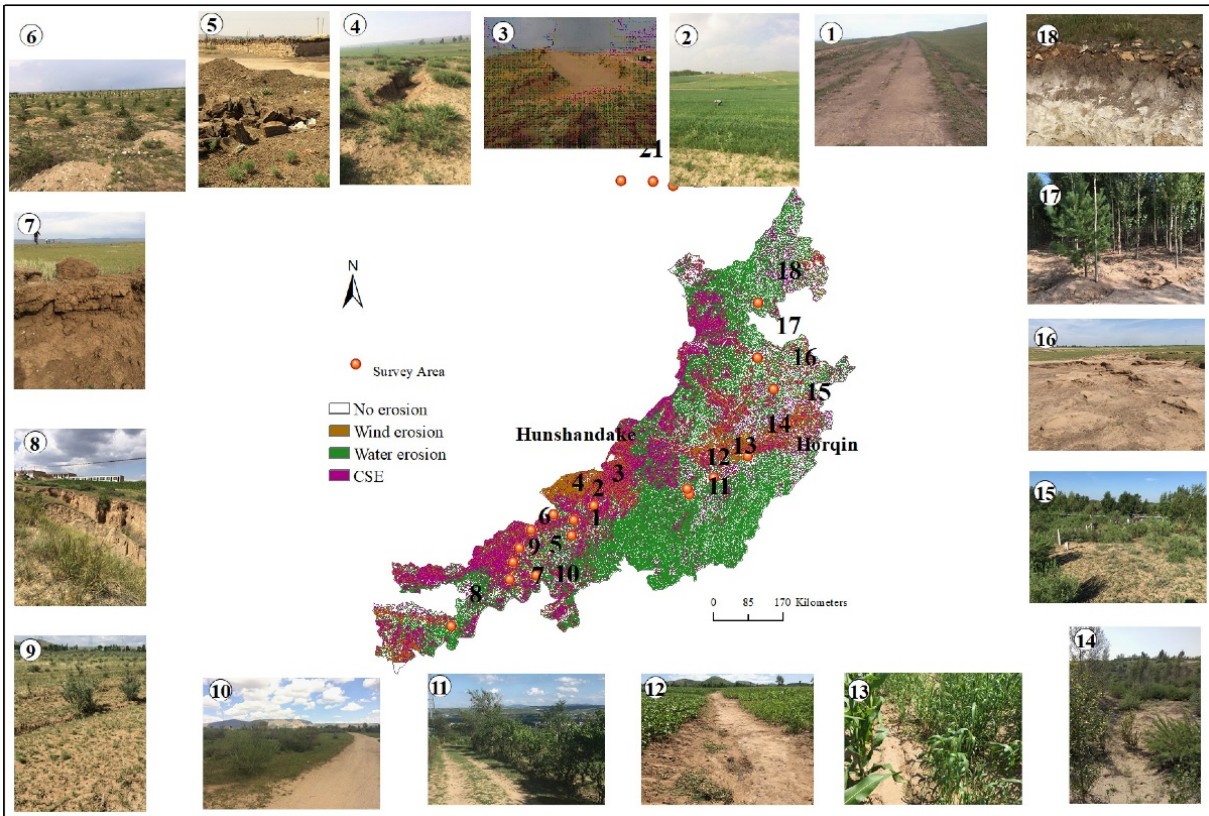

**Figure 8.** Photographs taken during ground-truthing showing typical soil erosion types in the study area (① the road effect by CSE in Taipusi Banner (County); ② seasonal bare sloping farmland affected by CSE in Taipusi Banner (County); ③ the sand covered loess ditch formed by CSE in Zhenglan Banner (County), for which a ditch is affected by water erosion and the sand is covered by wind in winter and spring; ④ ditch formed by water erosion in Huade County; ⑤ loess and sand road affected by CSE and human activity in Huade County; ⑥ loess and sand mound affected by CSE and mouse in Shangdu County; ⑦ loess and sand road affected by CSE in sand covered loess area in Qahar YouyiHou Banner (County); ⑧ water erosion ditch affected by water erosion in Jungar Banner (County); ⑨ medium coverage slope grassland affected by water erosion in Qahar Youyi Qian Banner (County); ⑩ the road covered by sand effect by CSE in Xinghe County; ⑪ ditch affected by water erosion in Hongshan District, Chifeng City; ⑫ water-eroded ditch on sloping farmland in Songshan District, Chifeng City; ⑬ corn field covered with sand affected by wind erosion in Aohan Banner (County); ⑭ low-cover dune land affected by wind erosion in Naiman Banner (County); ⑮ medium-covered dune land affected by wind erosion in Horqin Zuoyinghou Banner (County); ⑯ the bare low-cover grass destroyed by CSE in Horqin Zuoyi Zhong Banner (County); ⑰ economic forest without vegetation under the forest affected by water erosion in Horqin Youyi Zhong Banner (County); ⑱ the source-bordering dune, the result from aeolian–fluvial interaction; one of the main requirements for which is a regular source of sand from a seasonally flowing sand-bed channel in Horqin Right Front Banner (County)).

**Table 6.** Accuracy of identifying the distribution of soil erosion types.

| | | Ground-Truthing | | | | | |
| --- | --- | --- | --- | --- | --- | --- | --- |
| | | No Erosion | Water Erosion | Wind Erosion | CSE | Total | User Accuracy (%) |
| | No erosion | 0 | 1 | 0 | 2 | 3 | 0 |
| Identifying the | Water erosion | 0 | 6 | 0 | 0 | 6 | 100 |
| soil erosion | Wind erosion | 0 | 0 | 3 | 0 | 3 | 100 |
| type results | CSE | 2 | 0 | 0 | 4 | 6 | 66.67 |
| | Total | 2 | 7 | 3 | 6 | 18 | 72.22 |

## 4. Discussion

### 4.1. Effectiveness and Seasonality of CSE

It is apparent from our results (Figure 4) that CSE is most closely associated with the wind erosion and water erosion transition zone, i.e., the agro–pastoral ecotone of northern China. Typical CSE landforms are formed when a water eroded gully is infilled by sandy loess during the drier winter and spring months, or where a water-eroded gully forms during the summer monsoon season and, in both instances, this elevates the overall rate of erosion (Figure 9a,b,e,f). Water-eroded gullies also develop from wind-eroded depressions in the sand-covered loess area on windward slopes in the study area. Moreover, water erosion under high rainfall intensities during the summer monsoon results in exposure of material at the surface that is relatively easily eroded by wind in the following winter and spring, again thereby increasing total soil erosion (Figure 9c,d). Such phenomena are common in the part of the study area where CSE is most prominent.

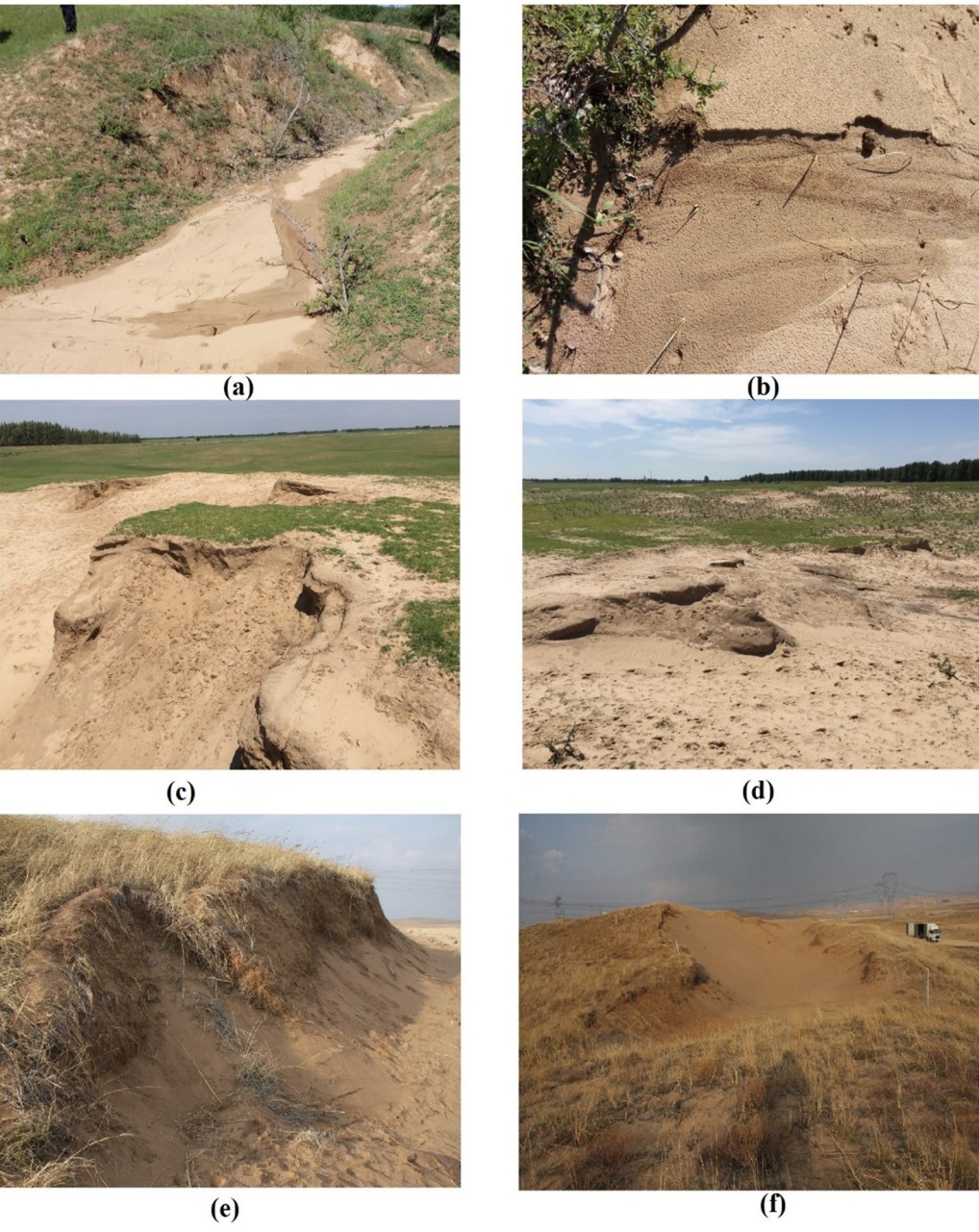

**Figure 9.** Photographs of typical CSE landforms in the study area ((**a,b,e,f**): water erosion gully filled by sand and loess; (**c,d**): water erosion gully developed from wind erosion scour).

The underlying driving forces of CSE are rainfall and wind erosivity acting in concert with surface conditions. The combination of such forces results in wind and water compound erosion (WWCE) and wind and water sequential erosion (WWSE). In order to resolve these processes further, we first calculated the mean monthly rainfall erosivity (Figure 10a) and wind erosivity (Figure 10b) from 1975 to 2015, followed by computing the combined effects of each force (Figure 11).

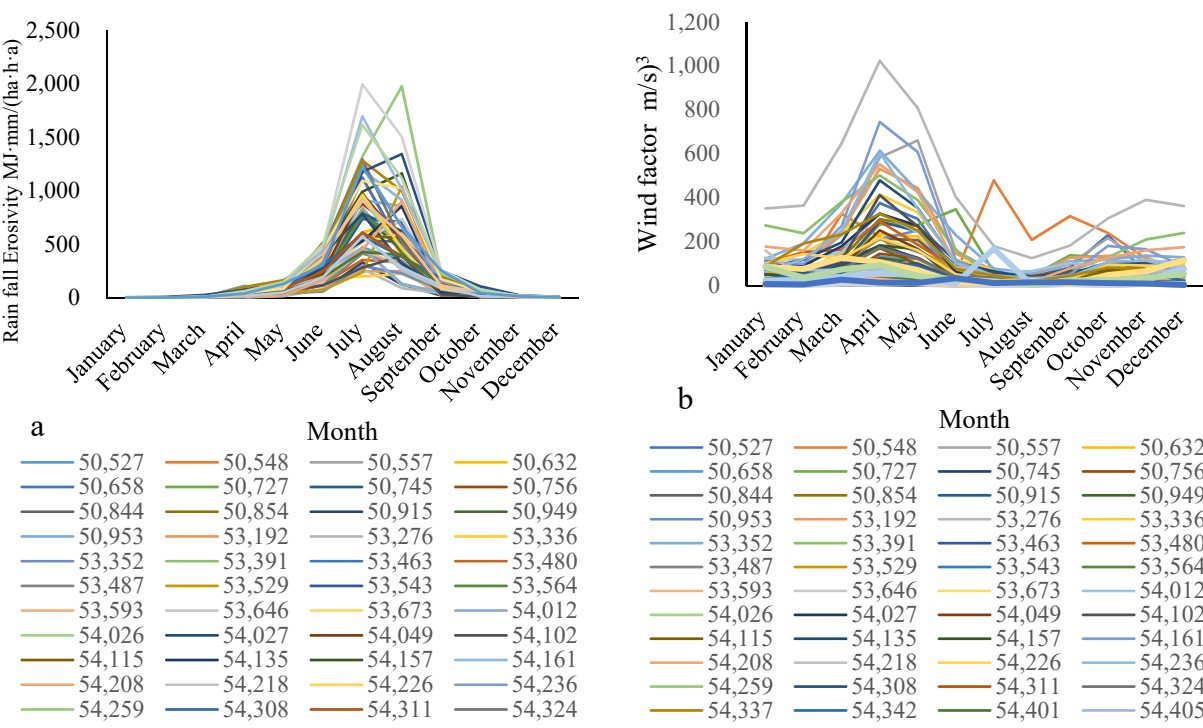

**Figure 10.** Mean monthly rainfall erosivity and wind factors in the study area (1975–2015) ((**a**): rainfall erosivity; (**b**): wind erosivity factor).

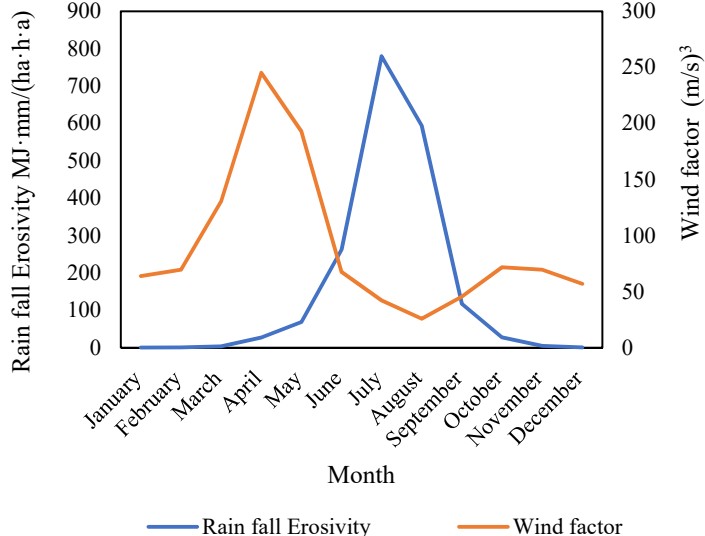

**Figure 11.** Graph showing the monthly sequence of water and wind factors in the study area.

Figures 10 and 11 illustrate that rainfall erosivity is higher during the summer months from May to August, with a maximum monthly rainfall erosivity value of approximately 800 MJ mm ha$^{-1}$ h$^{-1}$ in July, while wind factors are higher during late winter and spring months, from March to May, with a maximum monthly wind factor value of approximately

700 (m/s)$^3$ in April. A marked seasonal pattern is therefore evident whereby wind erosion is dominant from January to May and October to December, and water erosion from July and August; CSE is most likely during the transition months of June and September.

The efficacy of CSE in increasing soil erosion has been demonstrated experimentally. Yang et al. [28] used a wind tunnel and rainfall simulator under laboratory conditions to explore the combined effects of water erosion and subsequent wind erosion on a bed surface. Firstly, the influence of wind erosion on subsequent water erosion was examined, simulating conditions that are mainly associated with water-eroded gullies in the study area. Elevated wind speeds were shown to clearly alter the microtopography of the gullies, and extended the gully walls laterally, thereby exacerbating water erosion on the bed surface [28]. Yang et al. [29] then studied the effect of water erosion on a wind-eroded surface. In this case, water erosion had the effect of suppressing wind erosion due to the formation of a lag layer following rainfall [29], an observation further supported by Draut's [58] study that indicated that aeolian processes were effective locally in counteracting gully erosion.

### 4.2. The Contribution of This Study to Soil Erosion Research

Previous studies of CSE have been conducted at a large spatial scale, and the existence of mixed pixels has made it difficult to assign them accurately to CSE. The use of higher resolution (30 m) pixel data in this study, however, assists in resolving this such that a more accurate representation of the distribution and rate of soil erosion in the study area is possible. According to the National Soil Erosion Classification Grading Standard, we present here a new soil-erosion map of the study area at much greater spatial resolution (Figure 12). The identification and assessment of CSE reveals a higher rate of soil erosion in general, as well as greater prominence of strong and extremely strong erosion in particular. Water erosion dominates in the central and southern mountains, while slight water erosion is distributed mainly in the northeastern parts of the study area.

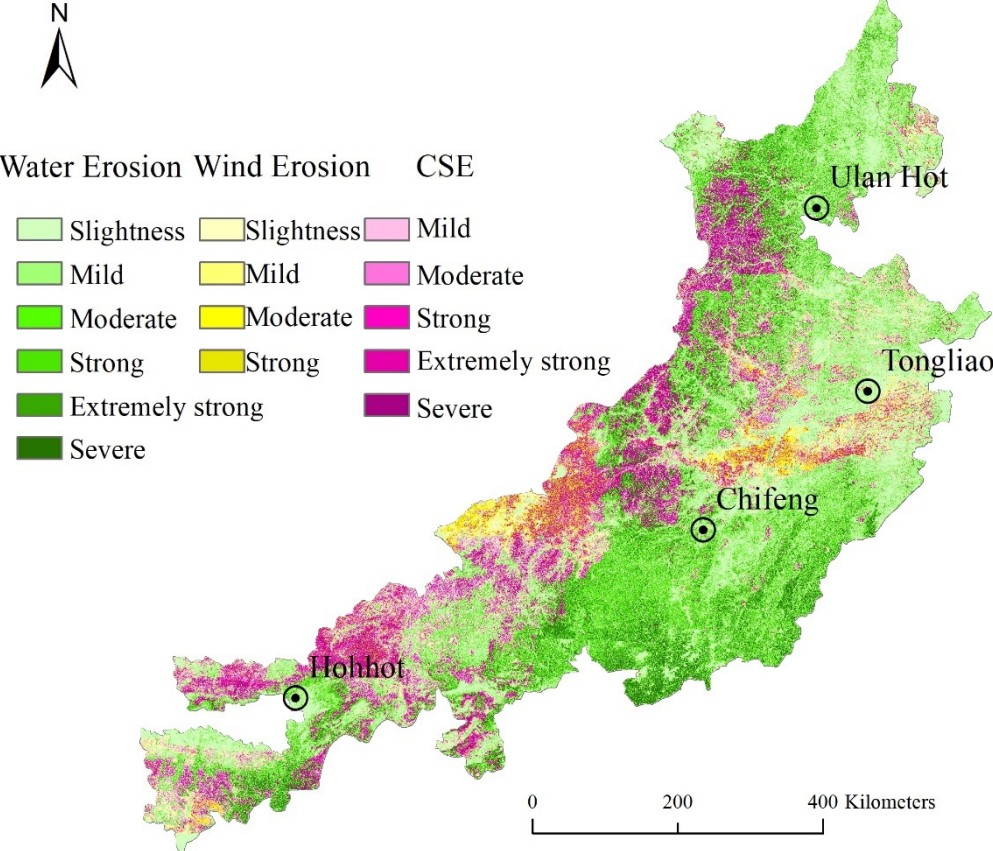

**Figure 12.** Distribution of soil erosion magnitude classes in the study area (1975–2015).

*4.3. Uncertainty of Effects of Wind-on-Water and Water-on-Wind*

This paper reports that the influences of wind on water and water on wind can either augment or suppress total erosion, depending on the circumstances [28,29]. This uncertainty is in part due to differences in the spatial scale of the model, and of field and laboratory observed data. For example, the wind-on-water and water-on-wind experimental results of the slope mechanism experiment are not suitable for upscaling [28]. However, it is evident from laboratory studies that wind erosion may amplify water erosion, and that water erosion may reduce subsequent wind erosion [28,29], indicating that wind on water and water on wind can offset each other. Furthermore, after exploring the combination of external wind- and water-erosion factors using weather station data, it is notable that most of the weather stations should be characterized by wind and water sequential erosion (WWSE), suggesting that wind and water erosion become superimposed in space. Moreover, the months associated with high rainfall erosivity values and high wind factor values do not generally overlap in time, since wind erosion plays the dominant role from January to May and again from October to December, while water erosion is an important force from July to August. Overall, this suggests that the influence of wind-on-water and water-on-wind would most likely occur in the months of June and September.

Nevertheless, uncertainties remain. Firstly, it is not clear if the influence of wind-on-water and water-on-wind erosion may be offset. Secondly, when the sediments deposited by wind erosion enter into gullies, the kinetic energy required for water erosion would increase [59], which is an issue that warrants further investigation. Thirdly, Mohammed et al. [60,61] found that all factors (i.e., inclination, rainfall, and land use) had a significant ($p < 0.001$) effect on soil loss, and that human activities, such as land use [61] and the selection of vegetation type [62], play an effective role in mitigating soil-erosion rates. Further research on the degree to which wind-to-water and water-to-wind may suppress or accentuate erosion is also needed under human activities. In the present study, there are constraints due to spatial and temporal resolution of some data; for example, while the main influencing factors (topography, rainfall erosivity, and wind factors) for CSE-erod are available at 30 m resolution, other data are only available at a coarser scale, meaning that the calculated soil-erosion rates must be considered as estimates only.

## 5. Conclusions

This paper uses the Chinese Soil Loss Equation (CSLE) and the Revised Wind Erosion Equation (RWEQ) to calculate the rate of soil erosion and map the distribution of three types of soil erosion classified as (i) wind (wind-erod), (ii) water (water-erod), and (iii) CSE (CSE-erod) for a study area that spans more than 400,000 km$^2$ of sand- and loess-covered landscapes in northern China. The main conclusions are as follows: Firstly, according to minimum threshold values for mild erosion, we identify water-erod, wind-erod, and CSE-erod land as occurring across 41.41%, 13.39%, and 27.69% of the total area, while mean soil erosion rates for water-erod, wind-erod, and CSE-erod land were calculated as 6877.65 t km$^{-2}$ yr$^{-1}$, 1481.47 t km$^{-2}$ yr$^{-1}$, and 5989.49 t km$^{-2}$ yr$^{-1}$, respectively. Secondly, land subject to CSE-erod is distributed especially around the margins of those areas that experience wind erosion and water erosion independently, and the erosion area is 20.26% larger than when considering water erosion alone, and 48.18% larger than when considering wind erosion alone, while the erosion rate is 11.03% higher than in the case of water erosion alone, and 88.97% higher for wind erosion alone. Thirdly, CSE-erod is an important but underappreciated process in semiarid regions, and it needs to be accounted for in land degradation assessments, as it has substantial impacts on agricultural productivity and sustainable development in regions with sandy and/or loess-covered surfaces.

**Author Contributions:** Conceptualization, P.S. and J.W.; methodology, D.L.; software, D.L.; validation, D.L., M.M. and H.Y.; formal analysis, M.M.; investigation, P.S., G.Z. and H.Y.; writing—original draft preparation, D.L.; writing—review and editing, M.M.; supervision, Z.H.; project administration, G.Z.; funding acquisition, D.L. All authors have read and agreed to the published version of the manuscript.

**Funding:** This study was funded by the National Natural Science Foundation of China (Project No.: 41271286), the National Key Research and Development Program (No. 2016YFA0602402), and the Natural Science Foundation of Zhejiang Province, China (LQ18D010003).

**Institutional Review Board Statement:** Not applicable.

**Informed Consent Statement:** Not applicable.

**Data Availability Statement:** No new data were created or analyzed in this study. Data sharing is not applicable to this article.

**Conflicts of Interest:** The authors declare no conflict of interest.

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
