# Peer review of "Measuring Compound Soil Erosion by Wind and Water in the Eastern Agro–Pastoral Ecotone of Northern China"

_sustainability, doi:10.3390/su14106272_

Round 1

Reviewer 1 Report

This is an interesting and well prepared manuscript. 

There are only a few points that I would ask the authors to consider:

- Introduction: Please add more information and references to the models used (CSLE and RWEQ).

- Table 3 and Table 4: Please adjust the table layout and explain all abbreviations in the table footers

- Material and methods: I miss a description of the field investigations mentioned (some results of the field investigations are listed in section 3.5). Please add a subsection for this.

- Section 2.3.6: Please add some information on how the validation maps (CAS and BNU) are produced (in principle).

- Section 3.5: This chapter is quite short, please add more information as validation is very important.

Author Response

This is an interesting and well prepared manuscript. 

There are only a few points that I would ask the authors to consider:

R1: Introduction: Please add more information and references to the models used (CSLE and RWEQ).

A: Thanks for the comment. We have added more information and references to the models used (CSLE and RWEQ). CSLE has been developed for estimating annual soil erosion by water in China. RWEQ has been developed as a tool for identifying practices that control windblown soil loss.

Please see lines 94 to 99.

R2: Table 3 and Table 4: Please adjust the table layout and explain all abbreviations in the table footers.

A: Thanks for the comment. We have adjusted the table layout and explained all abbreviations in Tables 3 and 4.

Please see lines 171 to 193, and lines 212 to 227.

R3: Material and methods: I miss a description of the field investigations mentioned (some results of the field investigations are listed in section 3.5). Please add a subsection for this.

A: Thanks for the comment. We have added text describing the methods used in field investigations.

Please see lines 250 to 255.

R4: Section 2.3.6: Please add some information on how the validation maps (CAS and BNU) are produced (in principle).

A: Thanks for the comment. We constructed a scatter plot with 1:1 line and data distribution of water erosion magnitude comparing the CAS and BNU outputs with this study (Figures 5 and 6) and then compared wind erosion magnitude between CAS and this study using the same method (Figure 7).  But the problem you have then is that these Figures need to be cited before these. So consider moving this comment to the validation section

We have added explanatory text accordingly.

Please see lines 288 to 290.

R5: Section 3.5: This chapter is quite short, please add more information as validation is very important.

A: Thanks for the comment. We have added more information about typical examples of CSE-erod in Section 3.5.

Please see lines 360 to 364.

Reviewer 2 Report

Dear Authors, Please make minor revisions taking into account some of the explanations I have added on MS. 

Author Response

Reviewing: 2

R1: Please add year of this reference or use numerical notation. Check!

A: Thanks for the comment. We checked and revised references throughout the manuscript.

R2: Please add UTM coordinates.

A: Thanks for the comment. We have included coordinates in the revised manuscript and in Figure 1.

Please see lines 114 to 115, and Figure 1.

R3: If possible, please add soil information.

A: Thanks for the comment. We have added some information about the soil characteristics.

Please see lines 131 to 133.

R4: Why Kappa? Please explain.

A: Thanks for the comment. We explain it in the revised manuscript. The Kappa coefficient is proposed here for assessing the accuracy of classifiers. Cohen’s Kappa statistic compensates for classifications that may be due to chance and is a widely used method in these circumstances.

Please see lines 260 to 262.

We appreciate for Reviewer’s warm work earnestly, and hope that the corrections will meet with the editor’s approval.

Once again, thank you very much for your comments and suggestions.

Reviewer 3 Report

Manuscript from Sustainability– MDPI

Measuring compound soil erosion by wind and water in the eastern agro-pastoral ecotone of Northern China

Major Comments:

The current study focuses on calculate the rate of soil erosion and map the distribution of three types of soil erosion  as: wind (wind-erod), water (water-erod) and compound soil erosion (CSE-erod) for the study area that spans sand- and loess-covered northern China based on the Chinese Soil Loss Equation (CSLE) and the Revised Wind Erosion Equation (RWEQ). The authors undertake a key task to identify the three types of soil erosion associated with actual mechanisms controlling the interactions between wind and water erosion which are substantial impacts on agricultural productivity and sustainable development in semi-arid regions. After tracking all sides within the manuscript, the author used a geo-analysis to assess the distribution of three types of soil erosion. I think the paper does not need a big change it was very interesting and sound and in line with the scientific overview of the journal scope.

- I think the manuscript does not need a comprehensive change. The paper has no limitations; but I have some key notes

L54, please add a clear reference here in numbers.

L83, and L85, revise the citation to Tuo et al.  [25] and Yang et al. [26]

L144, please add a clear reference here, also within the references list for Liu et al. (2002) in numbers

L151, The soil loss's unit is expressed in tonnes per hectare per year, but you mapped it as tonnes per km-2 per year, please justify this treatment.

L54, please add a clear reference here in numbers

L167, the description of the elements of the Chinese Soil Loss Equation (CSLE) formulas should be add in details. Which you have presented in Table 3. These characterizations of the elements are extremely important to lay readers, to avoid returning to the studies referred to.

L171, please add a clear reference here in numbers and including it within the list

L176, SL in kg m-2: How did you convert to  t km-2 yr-1?

L184, As the previous comment, the description of the elements of the Revised Wind Erosion Equation (RWEQ) formulas should be add in details. Which you have presented in Table 4.

L 202, do you mean that the T2 is the length of time of water impact on wind erosion, not the opposite? In this regard, how did you calculate the length of time (T) herein?

L 202 and 207, Please add a clear reference here in numbers, also add it within the references list for (Water and Soil Conservation Department, Ministry of Water Resources, 2008; Yuanyuan et al., 2016; Hua et al., 2019)

L 216,221, Please add a clear reference here in numbers, also add it within the references list

L 245, Do you mean: "According to the soil erosion intensity rating system of...> the Water and Soil Conservation Department of the Ministry of Water Resources [45]"

L 371 and 377, Some of references should be cited as: Yang et al. [26] and Yang et al. [27] as well. Please check all carefully based on the citation style of journal.

The discussion should be improved based on your finding, and addition other previous studies related the scope of the research from region in other countries, I think should be included and discussed, to comparing with the current results. see https://doi.org/10.3390/w12102786 and  https://doi.org/10.1111/sum.12683  

Author Response

Reviewing: 3

Major Comments:

The current study focuses on calculate the rate of soil erosion and map the distribution of three types of soil erosion  as: wind (wind-erod), water (water-erod) and compound soil erosion (CSE-erod) for the study area that spans sand- and loess-covered northern China based on the Chinese Soil Loss Equation (CSLE) and the Revised Wind Erosion Equation (RWEQ). The authors undertake a key task to identify the three types of soil erosion associated with actual mechanisms controlling the interactions between wind and water erosion which are substantial impacts on agricultural productivity and sustainable development in semi-arid regions. After tracking all sides within the manuscript, the author used a geo-analysis to assess the distribution of three types of soil erosion. I think the paper does not need a big change it was very interesting and sound and in line with the scientific overview of the journal scope.

- I think the manuscript does not need a comprehensive change. The paper has no limitations; but I have some key notes

R1: L54, please add a clear reference here in numbers.

A: Thanks for the comment. We have included a reference as suggested.

Please see line 55.

R2: L83, and L85, revise the citation to Tuo et al.  [25] and Yang et al. [26]

A: Thanks for the comment. We have revised these references accordingly.

Please see lines 84 to 86.

R3: L144, please add a clear reference here, also within the references list for Liu et al. (2002) in numbers

A: Thanks for the comment. We have added a reference as suggested and responded accordingly to your suggestion regarding Liu et al.

Please see line 147.

R4: L151, The soil loss's unit is expressed in tonnes per hectare per year, but you mapped it as tonnes per km-2 per year, please justify this treatment.

A: Thanks for the comment. A is expressed in tonnes per square kilometer per year, t km-2 yr-1. We have revised it.

Please see line 154.

R5: L54, please add a clear reference here in numbers

A: Thanks for the comment. We have now included a reference as suggested

Please see line 160.

R6: L167, the description of the elements of the Chinese Soil Loss Equation (CSLE) formulas should be add in details. Which you have presented in Table 3. These characterizations of the elements are extremely important to lay readers, to avoid returning to the studies referred to.

A: Thanks for the comment. We have explained all abbreviations in Tables 3 and 4.

Please see lines 171 to 193.

R7: L171, please add a clear reference here in numbers and including it within the list

A: Thanks for the comment. We have included a reference.

Please see line 197.

R8: L176, SL in kg m-2: How did you convert to t km-2 yr-1?

A: Thanks for the comment. SL is soil erosion per unit area (kg m-2), and 1 kg m-2 convert to 1000t km-2 yr-1.

Please see lines 202.

R9: L184, As the previous comment, the description of the elements of the Revised Wind Erosion Equation (RWEQ) formulas should be add in details. Which you have presented in Table 4.

A: Thanks for the comment. We have explained all abbreviations after Table 4.

Please see lines 212 to 227.

R10: L 202, do you mean that the T2 is the length of time of water impact on wind erosion, not the opposite? In this regard, how did you calculate the length of time (T) herein?

A: Thanks for the comment. According to the wind factor (Wf) and rainfall erosivity factor (R), T1 is the length of time of wind impact on water erosion, usually occurring in June (See section 4.1). T2 is the length of time of water impact on wind erosion, and usually occurs September (See section 4.1). T3 is the time of sequential erosion.

Please see lines 244 to 247.

R11: L 202 and 207, Please add a clear reference here in numbers, also add it within the references list for (Water and Soil Conservation Department, Ministry of Water Resources, 2008; Yuanyuan et al., 2016; Hua et al., 2019)

A: Thanks for the comment. We have included these additional references.

Please see line 250.

R12: L 216,221, Please add a clear reference here in numbers, also add it within the references list

A: Thanks for the comment. We have included a clear reference as suggested.

Please see lines 259 to 262.

R13: L 245, Do you mean: "According to the soil erosion intensity rating system of...> the Water and Soil Conservation Department of the Ministry of Water Resources [45]"

A: Thanks for the comment. Yes, we have revised the phrase in the manuscript.

Please see lines 294 to 296.

R14: L 371 and 377, Some of references should be cited as: Yang et al. [26] and Yang et al. [27] as well. Please check all carefully based on the citation style of journal.

A: Thanks for the comment. We have revised these references accordingly.

Please see lines 427 to 435.

R15: The discussion should be improved based on your finding, and addition other previous studies related the scope of the research from region in other countries, I think should be included and discussed, to comparing with the current results. see https://doi.org/10.3390/w12102786 and  https://doi.org/10.1111/sum.12683  

A: Thanks for the comment. We have included additional discussion in in section 4.3. Mohammed et al found that all factors (i.e., inclination, rainfall and land use) had a significant (p < 0.001) effect on soil loss, and that human activities, such as land use, the selection of vegetation type, play an effective role in mitigating soil erosion rates. Further research on the degree to which wind-to-water and water-to-wind may suppress or accentuate erosion is also needed in the context of human activities.

Please see lines 471 to 474.
